# Gait Characteristics during Dual-Task Walking in Elderly Subjects of Different Ages

**DOI:** 10.3390/brainsci14020148

**Published:** 2024-01-31

**Authors:** Nenad Nedović, Fadilj Eminović, Vladana Marković, Iva Stanković, Saša Radovanović

**Affiliations:** 1College of Health Sciences, Academy of Applied Studies Belgrade, 11000 Belgrade, Serbia; nedovicn@gmail.com; 2Faculty of Special Education and Rehabilitation, University of Belgrade, 11000 Belgrade, Serbia; eminovic73@gmail.com; 3Neurology Clinic, Clinical Center of Serbia, Faculty of Medicine, University of Belgrade, 11000 Belgrade, Serbia; vladanaspica@gmail.com (V.M.); idstanko139@gmail.com (I.S.); 4Institute for Medical Research, University of Belgrade, 11000 Belgrade, Serbia

**Keywords:** healthy elderly, aging, gait cycle, stride cycle, dual task, multiple task

## Abstract

Background: In older age, walking ability gradually decreases due to factors including impaired balance, reduced muscle strength, and impaired vision and proprioception. Further, cognitive functions play a key role during walking and gradually decline with age. There is greater variability in gait parameters when the demands during walking increase, in dual- and multiple-task situations. The aim of this study was to analyze gait parameters while performing a demanding cognitive and motor dual task in three different age-related healthy elderly subject groups. Method: A total of 132 healthy individuals (54 males, 78 females) were divided into three groups—55 to 65, 66 to 75, and 76 to 85 years. The subjects performed a basic walking task, dual motor task, dual mental task, and combined motor and mental task while walking. The gait parameters cycle time, stride length, swing time, and double support time were noted, as well as the variability of those parameters. Results: Cycle time was longer and stride length was shorter in the >76-year-old group than in the 51–65-year-old group in all test conditions. A comparison of all three groups did not show a significant difference in swing time, while double support time was increased in the same group. Conclusions: Changes are observed when gait is performed simultaneously with an additional motor or cognitive task. Early detection of gait disorders can help identify elderly people at increased risk of falls. Employing a dual-task paradigm during gait assessment in healthy elderly subjects may help identify cognitive impairment early in the course of the disturbance.

## 1. Introduction

Gait is a complex motor activity of rhythmic movements of the lower limbs that provide support and propulsion for locomotion [1]. Gait consists of repeated steps building up a gait cycle. Gait cycle is defined as a distance starting with the contact of the heel of one foot with the ground and ending with the repeated contact of the same foot. Preserved gait performance and gait control, in addition to efficient integration of motor and sensory information, requires the active role of cognitive functions [2].

In older age, walking ability gradually decreases due to various factors including impaired balance, reduced muscle strength in the lower limbs, impaired vision and proprioception, and deterioration in general health and appearance of medical conditions [3,4,5]. The gait of the elderly is associated with greater variability and changes in certain characteristics compared with the gait of younger persons [3,6]. For example, walking speed correlates with general health and functional status; if the walking speed is lower, there is a higher risk of falls, hospitalization, and mortality and reduced quality of life [7].

Cognitive functions play a greater role during walking than previously assumed (e.g., [8]). With age, cognitive functions gradually decline [9]. The influence of cognitive impairment, specifically executive dysfunction and attention problems, on gait performance in healthy elderly subject groups has been reported (e.g., [8,10,11]). A large longitudinal study with more than 3000 healthy subjects showed that spatial and temporal changes in gait parameters were associated with cognitive decline in elderly individuals [12]. Poor executive functioning and processing speed were associated with increased double support time and step time variability, while memory was not associated with gait changes [4]. Even in young individuals, gait and its parameters (such as speed) are not automated and also require partial involvement of attention [13].

The association of gait with cognitive functions is particularly significant during dual-task walking [11,14,15]. There is greater variability when the demands during walking increase, e.g., in dual- and multiple-task situations [11,16]. Hollman and colleagues [16] reported that healthy older individuals walked more slowly with greater variability in stride velocity during the dual-task condition compared with middle-aged and young individuals, presumably due to impaired cognitive performance in the elderly.

The aim of the present study was to analyze gait parameters while performing a demanding cognitive and motor dual task in three different age-related healthy elderly subject groups.

## 2. Subjects and Methods

### 2.1. Subjects

In the period from May 2018 to December 2019, a total of 132 healthy individuals (54 males, 78 females) were recruited among the spouses, friends, and relatives of the patients at Neurology Clinic, Clinical Center of Serbia, Faculty of Medicine, University of Belgrade. Subjects were divided into three groups—from 55 to 65 years (54 subjects, mean age: 55.83 ± SD 4.17 years); between 66 and 75 years (54 subjects, mean age: 68.13 ± SD 2.69 years); and from 76 to 85 years (24 subjects, mean age: 81.04 ± SD 4.47 years). We encountered challenges in finding a suitable sample due to the comorbidities and conditions affecting the gait performance of elderly individuals. Individuals suffering from neurological disorders, orthopedic diseases, or other medical conditions that could impair their independent walking, as well as those that used a cane or walker, were not included in the study. All subjects were free of medication and were satisfied with their quality of life and residential status. The study was approved by the Ethical Committee of the Faculty of Medicine, University of Belgrade (approval number 2650/X-10, 10 April 2018), and written informed consent was obtained from each participant prior to their inclusion in this study. The study was performed in accordance with the ethical standards of the Declaration of Helsinki and its later amendments.

### 2.2. Experimental Protocol

Subjects performed a self-paced basic walking task, a dual motor task, a dual mental task, and a combined motor and mental task while walking [17]. The motor dual task comprised comfortable walking with a large glass fully filled with water, with the aim not to spill the water. The mental dual task was serial “7” subtraction while walking, starting from numbers 100, 95, 90, or 105, chosen randomly, whereas the combined motor and mental task required the subjects to perform serial subtractions as described above while walking with a glass filled with water. Prioritization was not given for this task. Measurements were performed using the GAITRite electronic walkway of 5.5 m active area (CIR Systems, Havertown, PA, USA). The data from the activated sensors were transferred to the computer. The software enables the calculation of temporospatial gait parameters. The data were exported for each task and each participant for further statistical analysis. Participants performed six passes, three times down the corridor and back, at their comfortable gait velocity, starting and ending their walks approximately 1–1.5 m before and after the walkway. On average, at least 44 steps per experimental condition and for each participant were performed (ranging between 44 and 48), which was sufficient to adequately assess gait variability according to a previous study [18]. The walking distance was approximately 50 m (six GAITRite passes × 8–9 m) for each given task (basic, motor, mental, and combined), amounting to 200 m of walking, which was sufficient to adequately assess gait variability.

### 2.3. Statistical Analysis

For each subject, we individually determined mean stride characteristics (cycle time—CT, stride length—SL, swing time—SWT, and double support time—DST) and measures of stride-to-stride variability. Stride-to-stride variability is presented as coefficient of variation (CV)—variability normalized to the mean value (CV = 100 × SD/mean) and expressed in percentages.

Mean stride characteristics and CV were statistically compared. ANOVA with post hoc Bonferroni correction was used to assess differences between the three investigated groups. Repeated measures ANOVA was used to calculate differences between parameters in different test conditions. Results of mean stride characteristics are presented as mean with standard deviation and results of coefficient of variation as mean. Results are depicted as *p*-values (significance *p* ≤ 0.05, Bonferroni correction). Statistical analysis was performed using SPSS 17.0 (SPSS, Chicago, IL, USA).

Means and standard deviations (SDs) for each stride characteristic (cycle time—CT, stride length—SL, swing time—SWT, and double support time—DST) and measures of stride-to-stride variability presented as coefficient of variation (CV) were converted to Cohen’s d effect size (ES). The calculated ES was interpreted using the conventions outlined for standardized mean difference: ES < 0.2—very small, 0.2 ≤ ES < 0.5—small, 0.5 ≤ ES < 0.8—moderate, ES ≥ 0.8—large.

## 3. Results

The comparison of the gait parameters and their CVs in the three subject groups during the performance of different walking tasks (base walk (B), walk with motor task (M), walk with mental task (7), and walk with combined task (C)) is presented in Figure 1, as well as in Table 1 and Table 2, while effect size values are presented in Table 3.

### 3.1. Cycle Time

Cycle time was significantly increased in the >76-year-old group in comparison to the 51–65-year-old group in all test conditions with a large effect size, while in the 66–75-year-old group it was increased only during the mental and combined tasks (Figure 1, Table 1 and Table 3). In contrast to the motor task, which did not influence cycle time in all the tested groups, the mental and combined tasks increased cycle time (Table 2). In the 51–65-year-old group, the combined tasks increased CV values. In the 66–75-year-old group, mental activity increased CV values compared with the base and motor conditions (Table 2).

### 3.2. Stride Length

Stride length was significantly shorter in the >76-year-old group than in the 51–65-year-old group in all test conditions, with a large effect size. However, in the 66–75-year-old group, only the mental and combined tasks shortened stride length in comparison to the youngest group, with a moderate effect size (Figure 1, Table 1 and Table 3). In all groups, motor, mental, and combined activity decreased stride length compared with stride length in basal conditions. Combined activity also significantly shortened stride length compared with motor and mental activity. Mental activity during gait shortened the stride length compared with motor activity only in the 66–75-year-old group. Mental and combined activity while walking significantly increased CV stride length in all subject groups, with a large effect size when comparing the youngest and middle groups (Table 2 and Table 3).

### 3.3. Swing Time

A comparison of all three groups did not show a statistically significant difference in swing time (Figure 1, Table 1). The mental and combined tasks increased swing time compared with the values in basal conditions in the 51–65-year-old and 66–75-year-old groups, while motor activity reduced swing time in the youngest group. The combined task increased swing time in the 51–65-year-old and 66–75-year-old groups. CV did not differ among the groups under different experimental conditions, except that the mental task increased CV swing time compared with motor task performance in the 66–75-year-old and >76-year-old groups (Table 2). Effect size was large when comparing the youngest and oldest groups in the mental and combined tasks (Table 3).

### 3.4. Double Support Time

Double support time was increased in the >76-year-old group compared with the 51–65-year-old group in all experimental conditions, with a large effect size, as well as compared with the 66–75-year-old group under motor and combined tasks (Figure 1, Table 1 and Table 3). Also, double support time was prolonged in the 66–75-year-old group compared with the 51–65-year-old group in base conditions, as well as under motor and mental tasks, with a moderate effect size (Table 3). The combined task performance increased double support time compared with values in base and motor task conditions in all groups. CV double support time did not differ in different experimental conditions (Table 2).

## 4. Discussion

Daily life activities require the simultaneous performance of multiple tasks, both motor and mental. With older age, the ability to divide attention and to perform multiple tasks simultaneously seems to be impaired [14,19]. In particular, when individuals have motor or cognitive impairments, it is more difficult for them to perform concurrent motor and cognitive tasks [20]. A hypothesis is that the two tasks influence each other and compete for cortical brain resources. Changes in movement, such as a slowing of movement speed, which are often defined as “dual task cost”, result from the involvement of cortical processes [11]. Gait changes which occur while performing another action can reveal impaired cognitive function in people with early Alzheimer’s or Parkinson’s disease. An impaired ability to divide cognitive resources between walking and an attention-demanding task put the patients at increased risk of falling [21]. Some changes in the walking mechanism, such as a reduction in step length and walking speed, while at the same time prolonging the phase of support on both legs can be mechanisms of adaptation of older people to the possibility of experiencing a fall during walking [22].

Healthy young individuals also experienced slowing of gait as well as impaired performance of an additional task when they were asked to perform it while walking, suggesting that a complex activity such as walking relies on attention even in this population [23]. A mental task relies heavily on working memory capacity, and is thus directly dependent on executive functions [24]. Even in young individuals, gait (with its parameters such as speed) is an automated process, which also requires minimal involvement of attention [13].

In our study, gait cycle time significantly increased in the eldest subjects over 76 years of age compared with the youngest subjects between 51 and 65 years of age in all testing conditions. Compared with the subjects between 66 and 75 years of age, in the eldest group gait cycle increased during the performance of mental and combined tasks. CV was increased in the youngest group during the combined task and in the middle group during the mental task. Beauchet et al. [25] reported that the mean walking cycle duration in elderly people increased when they walked while simultaneously performing an arithmetic task (*p* = 0.005) but not a verbal fluency task, suggesting that gait speed during the dual task depends on the type of task presented to the subject. An increase in the CV of the gait cycle while performing the backward counting task may be related to the competition between the motor task (walking) and executive functions (counting). Also, an increase in the gait cycle time and variability was found in a similar study of elderly individuals when they were simultaneously performing a verbal fluency task and walking [26] or serially subtracting seven from an initial three-digit number [27].

The duration of swinging with one leg, defined as the time during which the observed foot is in the air and is not in contact with the ground (swing time), did not differ between subject groups in our study.

The stride length is the distance from the contact with the heel of one foot to the contact of the same foot with the ground again [28]. Stride length was significantly shorter in the eldest subject group (>76 years) compared with the youngest group (51–65 years) in all testing conditions. In the middle subject group (66–75 years old), only the mental and the combined tasks shortened the step length compared with the youngest group. In all groups, the additional task reduced step length compared with baseline gait, and in particular the combined task significantly reduced step length compared with the motor and cognitive tasks alone. The cognitive and combined tasks significantly increased the step length CV in all tested groups [29]. Dual-tasking, combining walking with a mental tracking task, affected spatiotemporal gait parameters with shorter stride length, longer stride time, and higher stride length and stride time variability during dual tasks in a group of elderly people [30].

Double support time increased in the subject group over 76 years of age compared with the group of 51–65 years of age during all test conditions, as well as compared with the group of 66–75 years of age during the motor and combined tasks. Also, the time of double support increased in the 66–75-year-old group compared with the 51–65-year-old group during the basic gait, as well as during the performance of the motor and cognitive tasks. The combined activity increased the time of double support in all groups compared with the values during the basic gait, as well as during the performance of the motor task. CV did not differ significantly during the specified test conditions. The duration of the gait cycle is associated with an increase in the duration of double support [31], which can be interpreted as helping to reduce the attentional demands during the swing phase and thus lower the risk of loss of balance. Changes in gait pattern while performing two tasks simultaneously may represent a strategy for maintaining gait efficiency [32,33].

Additionally, older individuals face increased attentional demands to sustain a stable gait, surpassing those of their younger counterparts [34]. As age progresses, cognitive resources decrease, contributing to diminished gait performance among older adults and elevating the likelihood of falls [16]. Observing complex cognitive tasks provides deeper insights into cognitive status [35], and the decline in gait parameters observed during dual-task activities serves as an early indicator of cognitive decline [36].

This study shows interesting and important results for the growing elderly population and provides ideas and future directions for new research in this area. One of the limitations of the study is that it did not investigate the long-term effects of dual-tasking on gait parameters, as it only measured them during the tasks themselves. Also, the study did not explore the specific mechanisms underlying the observed differences in gait parameters between groups.

## 5. Conclusions

With the increase in life expectancy, the number of elderly people in the population is increasing, and it is of great importance to examine them and detect disorders of various parameters in time. With aging, significant changes in gait parameters are observed, especially when gait is performed simultaneously with additional motor or cognitive tasks. The study findings suggest that future screening of individuals aged over 75 years can be conducted effectively through a straightforward dual-task analysis of gait. The insights gained from the study could contribute valuable information to the ongoing development of evaluation protocols and have the potential to shape existing rehabilitation procedures. Early detection of gait pattern changes can help identify elderly people at increased risk of falls and plan interventions to prevent negative consequences. Employing a dual-task paradigm during gait assessment in healthy elderly individuals may help identify those with a cognitive impairment effect on gait pattern early, which is of utmost importance for tailoring clinical care interventions and enrolling patients to clinical care and possible trials with disease-modifying drugs. Early correction of altered gait parameters will significantly improve the quality of life of the elderly population.

## Figures and Tables

**Figure 1 brainsci-14-00148-f001:**
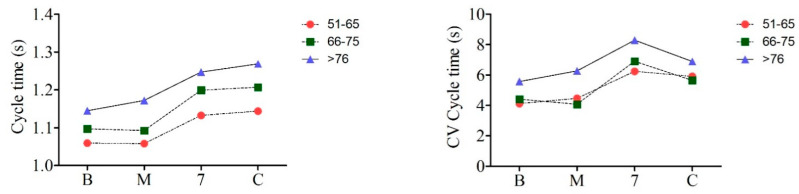
Comparison between three tested subject groups performing the following tasks: base walk (B), motor task (M), cognitive task (7), and combined motor and cognitive tasks (C). The four left panels represent cycle time (CT), stride length (SL), swing time (SWT), and double support time (DST), while the four right panels represent the CVs of CT, SL, SWT, and DST. Each symbol represents the mean values.

**Table 1 brainsci-14-00148-t001:** Cycle time (CT), stride length (SL), swing time (SWT), and double support time (DST)—mean values with SDs and CVs for the three tested groups performing the following walking tasks: base walk, motor task, mental task, and combined motor and mental tasks.

Groups	Base Walk			Motor Task			Mental Task			Combined Task		
	CT (mean)	±SD	CV (%)	CT (mean)	±SD	CV (%)	CT (mean)	±SD	CV (%)	CT (mean)	±SD	CV (%)
51–65	1.06	0.09	4.12	1.06	0.09	3.01	1.13	0.12	0.41	0.02	1.10	3.30
66–75	1.12	0.15	4.40	1.10	0.12	2.93	1.22 *	0.20	0.40	0.03 *	1.17	3.60
>76	1.16 **	0.12	5.57	1.18 **^,#^	0.10	5.82	1.26 **	0.13	1.19	0.03 **	1.20	3.11
*p* value	0.003		0.265	<0.001		0.352	0.02		0.371	0.02		0.649
	SL (mean)	±SD	CV (%)	SL (mean)	±SD	CV (%)	SL (mean)	±SD	CV (%)	SL (mean)	±SD	CV (%)
51–65	128.53	10.92	2.98	124.67	11.36	2.73	122.23	12.46	3.72	117.93	12.83	3.56
66–75	122.78	15.57	3.76	118.47	14.93	3.43 *	113.13 **	16.17	5.49 **	110.62 *	14.69	5.31 **
>76	112.02 **^,##^	19.39	4.39 **	104.49 **^,##^	15.70	4.35 **^,#^	99.65 **^,##^	17.31	6.77 **	100.27 **	14.07	6.34 **
*p* value	<0.001		0.003	<0.001		<0.001	<0.001		<0.001	<0.001		<0.001
	SWT (mean)	±SD	CV (%)	SWT (mean)	±SD	CV (%)	SWT (mean)	±SD	CV (%)	SWT (mean)	±SD	CV (%)
51–65	0.38	0.04	9.20	0.37	0.03	8.46	0.39	0.04	9.88	0.39	0.04	9.62
66–75	0.38	0.04	11.61	0.37	0.04	9.71	0.40	0.05	12.28	0.39	0.05	11.39
>76	0.38	0.03	12.36	0.37	0.03	10.85	0.39	0.04	15.91 **^,#^	0.39	0.05	15.01 **
*p* value	0.954		0.067	0.585		0.281	0.805		<0.001	0.945		0.01
	DST (mean)	±SD	CV (%)	DST (mean)	±SD	CV (%)	DST (mean)	±SD	CV (%)	DST (mean)	±SD	CV (%)
51–65	0.33	0.09	18.37	0.33	0.09	20.85	0.36	0.10	22.61	0.38	0.11	19.59
66–75	0.39 *	0.14	18.40	0.37 *	0.08	15.40	0.44 **	0.13	22.47	0.43	0.09	18.26
>76	0.43 **	0.10	26.42	0.44 **^,##^	0.09	15.34	0.50 **	0.10	29.68	0.58 **^,##^	0.23	29.57
*p* value	<0.001		0.211	<0.001		0.195	<0.001		0.341	<0.001		0.155

*p* < 0.05 * vs. group 51–65, ^#^ vs. group 66–75; *p* < 0.01 ** vs. group 51–65, ^##^ vs. group 66–75.

**Table 2 brainsci-14-00148-t002:** Differences between tasks (base walk, motor, mental, and combined task) are presented separately for the three tested groups.

	Groups	Motor vs. Base	Mental vs. Base	Combined vs. Base	Motor vs. Mental	Combined vs. Motor	Combined vs. Mental
CT	51–65	1.000	<0.001	<0.001	<0.001	<0.001	0.631
	66–75	1.000	<0.001	<0.001	<0.001	<0.001	1.000
	>76	0.827	0.005	0.001	0.006	0.001	0.284
SL	51–65	<0.001	<0.001	<0.001	0.053	<0.001	<0.001
	66–75	<0.001	<0.001	<0.001	<0.001	<0.001	<0.001
	>76	0.020	0.001	<0.001	0.147	<0.001	0.006
SWT	51–65	0.009	<0.001	0.002	<0.001	<0.001	0.668
	66–75	0.238	<0.001	<0.001	<0.001	<0.001	1.000
	>76	1.000	0.165	0.234	0.041	0.087	1.000
DST	51–65	1.000	0.015	0.010	0.035	<0.001	1.000
	66–75	1.000	0.160	0.010	<0.001	<0.001	1.000
	>76	1.000	0.467	0.017	0.507	0.021	0.327
CV CT	51–65	1.000	0.098	0.014	0.499	0.759	1.000
	66–75	1.000	0.021	0.114	0.027	0.316	0.948
	>76	1.000	1.000	1.000	1.000	1.000	1.000
CV SL	51–65	0.994	0.008	0.029	<0.001	0.001	1.000
	66–75	1.000	<0.001	<0.001	<0.001	<0.001	1.000
	>76	1.000	<0.001	<0.001	0.001	0.001	1.000
CV SWT	51–65	1.000	1.000	1.000	0.781	1.000	1.000
	66–75	0.093	1.000	1.000	0.016	0.192	1.000
	>76	1.000	0.139	0.726	0.027	0.580	1.000
CV DST	51–65	1.000	1.000	1.000	1.000	1.000	1.000
	66–75	0.557	0.812	1.000	0.010	1.000	0.717
	>76	0.433	1.000	1.000	0.037	0.262	1.000

CT—cycle time, SL—stride length, SWT—swing time, DST—double support time. The results are depicted as *p*-values. Shaded fields indicate statistical significance (*p* < 0.05, Bonferroni correction).

**Table 3 brainsci-14-00148-t003:** Effect size (ES) values for cycle time (CT), stride length (SL), swing time (SWT), and double support time (DST) and CVs for the three tested groups performing the following walking tasks: base walk, motor task, mental task, and combined motor and mental task.

Base walk								
Groups	CT	SL	SWT	DST	CV CT	CV SL	CV SWT	CV DST
51–65 vs. 66–75	0.46	0.43	0.05	0.54	0.09	0.45	0.37	0.00
51–65 vs. >76	1.01	1.18	0.02	1.10	0.36	1.02	0.48	0.34
66–75 vs. >76	0.28	0.64	0.06	0.31	0.29	0.32	0.12	0.39
Motor task								
Groups	CT	SL	SWT	DST	CV CT	CV SL	CV SWT	CV DST
51–65 vs. 66–75	0.44	0.47	0.05	0.48	0.07	0.54	0.20	0.29
51–65 vs. >76	1.31	1.57	0.08	1.22	0.27	1.07	0.35	0.27
66–75 vs. >76	0.66	0.92	0.13	0.88	0.34	0.54	0.19	0.01
Mental task								
Groups	CT	SL	SWT	DST	CV CT	CV SL	CV SWT	CV DST
51–65 vs. 66–75	0.53	0.63	0.11	0.61	0.11	0.83	0.39	0.01
51–65 vs. >76	0.99	1.60	0.03	1.39	0.34	1.48	1.03	0.29
66–75 vs. >76	0.19	0.82	0.12	0.55	0.26	0.49	0.62	0.38
Combined task								
Groups	CT	SL	SWT	DST	CV CT	CV SL	CV SWT	CV DST
51–65 vs. 66–75	0.47	0.53	0.06	0.52	0.05	0.89	0.28	0.28
51–65 vs. >76	1.02	1.35	0.00	1.37	0.17	1.98	0.84	0.40
66–75 vs. >76	0.44	0.71	0.05	1.09	0.38	0.44	0.58	0.57

## Data Availability

The data presented in this study are available on request from the corresponding author. The data are not publicly available due to privacy concerns.

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
