# Peer review of "Gait Characteristics during Dual-Task Walking in Elderly Subjects of Different Ages"

_brainsci, 2024, doi:10.3390/brainsci14020148_

Round 1
Reviewer 1 Report (Previous Reviewer 1)
Comments and Suggestions for Authors
1. Authors are improved the review/literature.
2. Responded for previous comments in the paper.
3. Results and comparison tables are well furnished.
4. Authors, provide the future scope of the review with perfect area of research scopes.
Author Response
Dear Reviewer,
Thank you for your report. We are glad that you are satisfy with the changes and responses we made - improved literature, Result section and Tables. We added "and future scopes for new research in this area" in the Discussion, line 252.
Reviewer 2 Report (Previous Reviewer 2)
Comments and Suggestions for Authors
Interesting idea of this article. My recommendations are the following:
Line 9-13 I recommend mentioning that it is Background, not the purpose that is mentioned in lines 13-14.
I recommend entering the following words as keywords: dual task, multiple task.
Author Response
Dear Reviewer,
Thank you for your report. We accepted suggested changes - change in the sentence - Background, as well as to add two more keywords – dual task and multiple task. Therefore, we added keywords and changed the sentence.
This manuscript is a resubmission of an earlier submission. The following is a list of the peer review reports and author responses from that submission.
Round 1
Reviewer 1 Report
Comments and Suggestions for Authors
The authors analyzed the gait of the elderly in three different groups and concluded that the gait cycle time increased from 76 to 85. We appreciate their contributions, and the following comments were useful to the authors to enhance the quality of the work.
Comments
1. How were these gait data captured for this timing analysis?
Any computer vision technology or sensor method is used.
2. In this study, 132 elderly gait data were analyzed, and the authors did not mention how many days data have been captured, what constraints faced at the data capture time, and how it is overcome by authors.
3. The way the gait data were analyzed by any machine learning algorithm provided the information?
4. The height and weight of the participants were impacted by the cycling time of gait, and any such observation was evaluated.
5. How your conclusions are useful to the researchers, provide the applications for usage of your data. Provide your novelties.
Reviewer 2 Report
Comments and Suggestions for Authors
My recommendations are the following:
From the Experimental protocol subsection, moving the information about the coefficient of variation (CV) to the Statistical analysis section.
In the Results section, entering the tables in the test, it is easier to view the information.
In the Results section, you mention in the first sentence that Comparison of the gait parameters is presented in Figure 1, this is not found in the test, I recommend the correction. Also in the brackets of the targeted indicators, mention table 1 and fig. 1, I recommend the correction.
Discussion section, mention that - CV was increased in the youngest group during the combined task, and in the middle group during the mental task, this indicator to record an increase should be compared over time (TI-TF) and not between two tests motor. CV refers to the results recorded on each test. I recommend clarification. In this study, no effective intervention was presented, only a finding at a given time.
The tables do not show the data regarding the arithmetic averages on the basis of which the CV was calculated. I recommend the correction. Likewise, in the Statistical analyzes section, the arithmetic mean of the results is not mentioned. What is the limits of agreement for the CV. CV is calculated in percentages, and has some standard reporting intervals, it does not emerge from tables and interpretation. What statistical indicator does the first column in table 1 represent, on each test? I recommend the correction.
The bibliographic indexes mentioned 17, 19 for the CV refer to studies, not to statistical benchmarks. It is not clear what to mention these studies. I recommend possibly moving and adapting them in the Discussions section.
Page 1 last sentence, double the information, he also mentioned in the introduction what it represents - The duration of swinging with one leg, I recommend the correction.
I recommend calculating the sample power, to highlight the relevance of this study.
The statistical analysis is poorly defined. As a recommendation, it was interesting to see if there are differences between the sexes.
It is a constative study without any concrete intervention.
I recommend expanding the bibliography, the most recent index is from 2018. I also recommend expanding the Discussions section with concrete correlations between the results of this study and results from previous studies.
What are the limitations of the study? I recommend mentioning them at the end of the Discussions section.